# *Haematococcus pluvialis-*Derived Astaxanthin Is a Potential Neuroprotective Agent against Optic Nerve Ischemia

**DOI:** 10.3390/md18020085

**Published:** 2020-01-28

**Authors:** Wei-Ning Lin, Kishan Kapupara, Yao-Tseng Wen, Yi-Hsun Chen, I-Hong Pan, Rong-Kung Tsai

**Affiliations:** 1Department of Ophthalmology, Kaohsiung Medical University Hospital, Kaohsiung Medical University, Kaohsiung 807, Taiwan; ning1128@gmail.com; 2Graduate Institute of Clinical Medicine, Kaohsiung Medical University, Kaohsiung 807, Taiwan; 3Institute of Eye Research, Hualien Tzu Chi Hospital, Buddhist Tzu Chi Medical Foundation, Hualien 970, Taiwan; contactkishankapuara@gmail.com (K.K.); ytw193@gmail.com (Y.-T.W.); 4Biomedical Technology and Device Research Laboratories, Industrial Technology Research Institute, Hsinchu 310, Taiwan; minisatellites@itri.org.tw (Y.-H.C.);; 5Institute of Medical Sciences, Tzu Chi University, Hualien 970, Taiwan

**Keywords:** astaxanthin, Akt/mTOR, Nrf2, retinal ganglion cells, ischemia

## Abstract

Astaxanthin, a xanthophyll belonging to the family of carotenoids, is a potent antioxidant. However, much less is known about its protective effects on the oxidative stress of ischemic optic nerve. We hypothesized that astaxanthin treatment could protect retinal ganglion cells (RGCs) from death via anti-oxidative and anti-apoptotic responses. Adult male Wistar rats were fed astaxanthin (100 mg/kg/day) by daily gavage for seven consecutive days, either before or after inducing oxidative stress in the retina by photodynamic treatment. The visual function, RGC apoptosis, macrophage infiltration in the optic nerve, expression of p-Akt, p-mTOR, SGK1, pS6K, Nrf2, p62, TNFα, Il1β in retinas were investigated. The visual function and the RGC densities were significantly higher in both pre- and post-treatment groups. The numbers of apoptotic RGCs and extrinsic macrophage infiltration in the optic nerve were significantly decreased in both astaxanthin-treated groups. Furthermore, pre- and post-treatment of astaxanthin showed a higher expression of p-Akt, p-mTOR, Nrf2 and superoxide dismutase activity, and a lower expression of cleaved caspase-3, suggesting anti-apoptotic and anti-oxidative roles. Our findings indicate that astaxanthin can preserve visual function and reduce RGC apoptosis after ischemic insults. Including astaxanthin in daily diet as a supplement may be beneficiary for ischemic optic neuropathy.

## 1. Introduction

Non-arteritic anterior ischemic optic neuropathy (NAION) is the most common acute optic neuropathy in the population over 50 years of age. It is clinically characterized by acute, painless, unilateral visual loss, with color vision impairment and visual field defects. The estimated prevalence is 2.3–10.3 per 100,000 people both in the U.S. and in Taiwan [1,2]. Its pathogenesis is still unclear, except that it is known to be associated with some predisposing vascular risk factors, such as hypertension, diabetes mellitus, hyperlipidemia, nocturnal hypotension, smoking, sleep apnea and small optic disc [1,3,4]. So far, there is no effective treatment for NAION. It is supposed that NAION is caused by vascular insufficiency resulting from disturbed small-vessel autoregulation of posterior ciliary circulation that leads to optic nerve head (ONH) ischemia. One recent report demonstrated that the level of advanced oxidation protein products in plasma was higher in NAION patients than in healthy controls [5]. Cumulative oxidative stress may induce cellular damage, impairment of the DNA repair system, and mitochondrial dysfunction; all of these have been known as critical factors in acceleration of aging process and the development of neurodegenerative disorders [6,7]. In our previous study, activation of nuclear factor erythroid 2-related factor 2 (Nrf2)-antioxidant response element (ARE) signaling was demonstrated to promote retinal ganglion cell (RGC) survival and to prevent blood–optic nerve barrier disruption in a rat model of anterior ischemic optic neuropathy (rAION) [8]. Therefore, decreasing reactive oxygen species (ROS) production in ischemic optic neuropathy may prevent RGCs from death. 

Astaxanthin (astaxanthin; 3,30-dihydroxy-b,b-carotene-4,40-dione) is a commercially available health supplement, which belongs to a family of carotenoids. It is abundant in particular natural resources, such as salmon, trout, shrimp, and algae. Astaxanthin can decrease ROS-mediated cellular toxicity [9], and has been proven in the literature to possess anti-oxidative, anti-inflammatory, anti-diabetic, and anti-cancer effects [10,11,12,13,14,15,16]. In the eye, astaxanthin has demonstrated a protective effect on UV-induced photokeratitis via anti-oxidative, anti-inflammatory, and anti-apoptotic activity [17]. Astaxanthin also protects against retinal damage by inhibiting oxidative stress [18,19]. Furthermore, astaxanthin attenuates RGC death under various stresses in a murine model [17]. Oral supplementation of astaxanthin is also reported to enhance superoxide scavenging activity in human aqueous humor [20].

To date, no previous studies have investigated the effects of astaxanthin in optic nerve ischemia. The purpose of this study is to evaluate the neuroprotective effects of astaxanthin in an established rAION model.

## 2. Results

### 2.1. Astaxanthin Treatment Preserves Visual Function after rAION

The changes in flash visual-evoked potentials (FVEP) were measured after four weeks of rAION induction. Reduction in P_1_-N_2_ amplitude indicated a decrease of healthy optic nerve axons was stimulated. The mean amplitude of the P_1_-N_2_ waves in the sham, rAION, pretreatment, and post-treatment group were 66 ± 13 μV, 19 ± 4 μV, 48 ± 11 μV, and 36 ± 8 μV, respectively (Figure 1a,b). The amplitudes of the P_1_-N_2_ waves were preserved significantly in both the pre- and post-treatment group (*p* < 0.05, post hoc Dunn’s multiple comparisons test, n = 6 in each group), compared to the rAION group. However, there was no significant difference between the pre- and post-treatment groups.

### 2.2. Astaxanthin Treatment Increases RGC Survival and Reduces Apoptosis by mTOR/Akt Signaling Axis

Using retrograde fluoro-gold labeling, only RGC with intact axon are labeled. Hence, fluoro-gold labeled RGC density can be used to access RGCs that are protected from death. The central RGC densities in the sham, rAION, pretreatment, and post-treatment group were 1735 ± 185/mm^2^, 524 ± 174/mm^2^, 1411 ± 194/mm^2^, and 986 ± 215/mm^2^, respectively (Figure 2a,b). RGC survival was increased in both pre- and post-treatment groups by ~3 fold and ~2 fold, respectively (*p* < 0.05, post hoc Dunn’s multiple comparisons test, n = 6 in each group), compared to the rAION group.

The number of terminal deoxynucleotidyl transferase dUTP nick end labeling (TUNEL) positive RGCs of the sham, rAION, pretreatment, and post-treatment group retinas were 1.4 ± 0.5, 12.4 ± 2.2, 3.6 ± 1.9, and 6.2 ± 3.3, respectively (Figure 3a,b). Both pre- and post-treatment groups significantly reduced the number of apoptotic RGCs (*p* < 0.05, post hoc Dunn’s multiple comparisons test, n = 6 in each group), compared to the rAION group. Based on RGC densities and TUNEL assay, we concluded that RGC survival was higher in the pretreatment group compared to post-treatment group.

We found that levels of p-mTOR and its downstream targets of the mTORC1 pathway (S6K) and mTORC2 pathway (SGK1) were maintained compared to the sham group. Further, we found activation of Akt lowered the expression of cleaved caspase-3 in both pre- and post-treatment group, compared to rAION group (*p* < 0.05, post hoc Dunn’s multiple comparisons test, n = 4 in each group) (Figure 3c,d). These data suggest that astaxanthin exerts an anti-apoptotic effect.

### 2.3. Astaxanthin Treatment Halts Extrinsic Macrophage Infiltration in the Optic Nerve and Shows Anti-Inflammatory Potential in Retina

Four weeks after rAION induction, the ED-1 positive cells in sham, rAION, pretreatment, and post-treatment groups were 0.41 ± 0.51, 141.62 ± 30.09, 35.75 ± 18.74, and 68.55 ± 14.68, respectively (Figure 4a,b). There was a significant decrease in ED-1 positive cells in the optic nerve of both pre- and post-treatment groups compared to rAION. In addition, the ED-1 positive cells in pretreatment was significantly less than in the post-treatment group (*p* < 0.05, post hoc Dunn’s multiple comparisons test, n = 6 in each group). Further, the levels of Il1β and TNFα in the retina were significantly decreased in the pretreatment group (*p* < 0.05, post hoc Dunn’s multiple comparisons test, n = 4 in each group). However, we only observed significant changes in IL1β levels, but not in TNFα levels in the post-treatment group (p = 0.13) compared to the rAION group (Figure 4c,d). These data suggest that an anti-inflammatory potential of astaxanthin in pre-treatment may be better than post-treatment.

### 2.4. Astaxanthin Shows Anti-Oxidative Potential in Retina

Previous studies have shown that Nrf2-p62 regulates in a positive feedback loop [21]. We found increased levels of Nrf2, p62, and superoxidase dismutase (SOD) activity in both pre- and post-treatment groups (*p* < 0.05, post hoc Dunn’s multiple comparisons test, n = 4 in each group) (Figure 5a,b), compared to rAION group. This finding suggests that astaxanthin can exhibit a protective anti-oxidative effect.

## 3. Discussion

Our findings suggest that oral administration of astaxanthin, either via pre-treatment or post-treatment, is neuroprotective in rAION. mTOR/Akt/Nrf2 pathways were modulated by astaxanthin treatment against apoptosis and oxidative stress. However, neuroprotection is more remarkable in the pre-treatment group, which suggests the usage of astaxanthin as a dietary supplement before disease onset offers better protection. Astaxanthin is reported to have protective effects on RGC against various stresses in vitro [22]. Oral supplementation of astaxanthin significantly reduces retinal damage and alleviates the decrease in electroretinogram a- and b- wave amplitudes following retinal ischemia and reperfusion [18]. Consistent with previous reports, our findings state that treatment with astaxanthin reduced oxidative stress, RGC apoptosis, and inflammatory response to preserve the visual function in rAION.

The PI3K/Akt/mTOR pathway is an essential signaling pathway involved in cell growth and survival [23]. Moreover, recent reports also stated the involvement of the Akt signaling pathway in spinal cord injury after astaxanthin treatment [24]. In this study, we found that astaxanthin treatment maintained the levels of p-mTOR and its downstream factors of mTORC1 and C2 pathway after optic nerve ischemia. Moreover, astaxanthin treatment promoted Akt activation and decreased cleaved caspase-3 levels. Therefore, the anti-apoptotic effect of astaxanthin in rAION can be attributed to maintaining mTOR/Akt activation and preserving the RGCs against cell death.

Elevation in oxidative stress results in increased Nrf2 expression and its downstream detoxifying and AREs to alleviate oxidative stress [25]. In this study, the levels of Nrf2 and p62 were higher in the astaxanthin-treated groups compared with the rAION group, as it has been reported that p62 and Nrf2 regulate the expression in a positive feedback loop manner [21]. Moreover, higher levels of p62 will compete with Keap1 to bind with Nrf2, which further prevents the ubiquitination of Nrf2 [26]. The previous study also reported reduced retinal oxidative stress by astaxanthin treatment in diabetic rats [27]. Our findings suggested that the anti-oxidative effects of astaxanthin treatment might relate to the maintenance of the Nrf2-P62 positive feedback loop. 

Neuroinflammation has been shown to play a vital role after optic nerve ischemia [28,29,30,31]. In our observations, treatment with astaxanthin decreased the levels of IL-1β and TNFα in the retina and reduced ED1-positive macrophage infiltration in the optic nerve after rAION induction. Our previous reports have demonstrated that the vascular permeability of the optic nerve was highly increased within two days post AION induction and that vascular-borne macrophages would infiltrate into ON [29,32]. Our results suggested that astaxanthin can halt the ED-1(+) cells infiltration into the ON. Astaxanthin has been reported to exhibit anti-inflammatory effects in lipopolysaccharide-stimulated microglial cells, in which astaxanthin inhibited the expression of COX-2 and iNOS [33]. In another report, astaxanthin inhibited inflammation and reversed M1/M2 polarization of microglial cells via low-density lipoprotein receptor-related protein-1 (LRP-1) [34]. Previous reports demonstrated that ROS is involved in the blood–brain barrier disruption [35]. One study reported that astaxanthin treatment significantly attenuated brain edema and blood–brain barrier disruption in the experimental model of subarachnoid hemorrhage [36]. We also found a dramatic anti-inflammatory effect in the pretreatment group. We hypothesize that in the pretreatment group, the residual astaxanthin in the body may have already depleted ROS after rAION induction. Therefore, pretreatment with astaxanthin may reduce vascular-born macrophage infiltration in the optic nerve by preserving the vascular permeability, thus reducing the oxidative stress and neuroinflammation caused by increased vascular permeability after optic nerve ischemia.

Some important similarities between the rodent and clinical NAION are optic-nerve head edema, loss of retinal ganglion cells, no latency changes but decrease of amplitude in VEP responses and macrophage infiltration and microglia activation in the optic nerve [28,30,37]. However, the difference between human NAION and rAION is that rats have no systemic commodities, such as diabetes, hypertension and sleep apnea. In conclusion, we demonstrated that astaxanthin ameliorated the damages caused by oxidative and ischemic stress in rAION model via its anti-oxidative and anti-apoptotic effects in retina and anti-inflammatory effects in optic nerve and retina (Figure 6). Therefore, our findings from both retina and optic nerve provide a potential usage of astaxanthin in patients with disc at risk in NAION as the dietary astaxanthin supplement is safe and has no adverse side effects and excessive ROS production has been implicated in the pathogenesis of various eye diseases, including dry eye disease, corneal diseases, glaucoma, diabetic retinopathy, and age-related macular degeneration [38]. We suggest that astaxanthin can be used as a potent supplement for neuroprotection in clinical NAION. 

## 4. Methods

### 4.1. Animal Model and Ethics Statement

Adult male Wistar rats, weighing 150–180 g (7- to 8-weeks old), were obtained from the breeding colony of BioLASCO Co., Taiwan. Animal care and experimental procedures were performed in compliance with the Association of Research in Vision and Ophthalmology (ARVO) Statement for the Use of Animals in Ophthalmic and Vision Research. Moreover, the Institutional Animal Care and Use Committee (IACUC) at Tzu Chi General Hospital approved all of the animal experiments (approval number 108-01). All operations and surgeries were performed on animals under anesthesia, which was achieved by intramuscular administration of a ketamine (100 mg/kg body weight) and xylazine (10 mg/kg body weight; Sigma, St. Louis, MO, USA) cocktail. We generated rAION by photodynamic activation of rose Bengal. Photoactivation of rose Bengal produces free radical oxygen in the optic nerve head (ONH), creating an oxidative stress environment followed by ischemic optic neuropathy and RGC death [29,39]. The detailed procedure is mentioned in our previous reports [32,39]. In brief, after intravenous injection of rose Bengal, the ONH was exposed with a fundus lens, and photoactivation of rose Bengal was performed using green argon laser on ONH. Sham operation was performed using the same procedure without rose Bengal injection. Animal euthanasia was carried out by CO_2_ insufflation.

### 4.2. Preparation of Astaxanthin Extract from *Haematococcus pluvialis*


*Haematococcus pluvialis* microalgae were provided by the Taiwan Cement Corporation. The microalgae wall was disrupted with a high-pressure homogenizer at 1200 bar (APV-2000, APV Lab Series Homogenizers,SPX Flow Technology). Crushed microalgae solution was freeze-dried to powder, and astaxanthin was extracted with supercritical carbon dioxide from wall-disrupted microalgae. For astaxanthin extraction, 75 g dried *Haematococcus pluvialis* algae were loaded into the extraction vessel where the remaining top and bottom sections were trapped with glass beads. The extractor was placed in the heating chamber to maintain the operating temperature of 40 °C and pressure of 6526 psi. Extraction was carried out at a constant flow rate of 2–4 LPM up to 6 h. The extracted stream from the extractor was then depressurized through a pressure restrictor and the sample was collected at a 1 h interval in a brown bottle to prevent light-degradation. 

The astaxanthin content was analyzed by HPLC (Waters Alliance 2695 HPLC Separations Module) using a photodiode array detector and a YMC Carotenoid (4.6-mm × 25-cm, 5-μm) HPLC column. The injection volume of each sample was set to 20 μL. Three eluents were used: (A) Methanol and (B) t-butylmethylether and (C) phosphoric acid, 1% aqueous. The gradient of the mobile phases was 81% A, 15% B, 4% C at 0 min; 66% A, 30% B, 4% C at 15 min; 16% A, 80% B, 4% C at 23–27min; and 81% A, 15% B, 4% C at 27- 35 min. Astaxanthin was eluted at a rate of 1 mL/min and detected by absorbance at 474 nm.

### 4.3. Dosage Information and Sample Size Estimation for Treatment Groups

The animals were randomly assigned into four groups: sham, rAION, pretreatment, and post-treatment. Both pre- and post-treatment groups received a 100 mg/kg/day dosage of astaxanthin for seven days one week before or immediately after rAION (referred to as pre-treatment or post-treatment, respectively). We adopted the dosage from previous studies [40,41,42].

Based on our preliminary data and our previously published articles, we performed sample size estimation for one-way ANOVA, fixed effects using G*Power tool [43]. Our analysis suggests that the minimum total sample size in four groups (sham, rAION, pre-treatment and post-treatment) with 95% power is 16. Even though we expect an effect at sample size of 16, we used total sample size of 24. This will help ensure that we have enough power in case some of the pre- and post-treatment assumptions are not met.

### 4.4. Flash Visual-Evoked Potentials (FVEP)

The detailed procedure of recording FVEPs has been described in our previous reports [39,44,45]. In brief, we implanted electrodes on the occipital cortex region and used a visual electrodiagnostic system (E3 System, Diagnosys LLC, Lowell, MA, USA) to measure the FVEP. We compared the amplitude of the P1-N2 wave in each group to evaluate visual function (n = 6 rats per group).

### 4.5. Retrograde Labeling of RGCs with Fluoro-Gold

The detailed protocol of Fluoro-Gold labeling has been described in our previous reports [29,39,44,45,46,47,48,49]. Briefly, we performed retrograde labeling of the RGCs 1 week before the rats were euthanized. An amount of 2 μL of 5% Fluoro-Gold was injected into the superior colliculus on each side through a Hamilton syringe. One week after the labeling, the eyeballs were harvested. The retinal flat-mounts were examined for RGC densities in the central retina, defined by a 1-mm distance from the margin of the optic nerve head (n = 6 rats per group).

### 4.6. In Situ Terminal Deoxynucleotidyl Transferase dUTP Nick End Labeling (TUNEL) Assay

The eyes were fixed in 4%PFA and cryoprotected in 30% sucrose. Then, 20-μm-thick frozen section was obtained, and a TUNEL assay was performed according to the manufacturer’s instructions (DeadEnd Fluorometric TUNEL System, Promega) [50]. The frozen sections of the retina at a 1- to 2-mm distance from the ONH were chosen to ensure the use of equivalent fields for comparison. In all groups, ten random high-powered fields (HPF, 400×) were captured, and an average from three sections per retina was used for further analysis (n = 6 rats per group). We concluded TUNEL positive RGCs as apoptotic RGCs only if the TUNEL (green) signal overlapped with DAPI (blue). Only a green signal with no DAPI was considered as false positive and not counted as TUNEL positive RGCs.

### 4.7. Immunohistochemistry Staining of ED1-Positive Cells in the Optic Nerve

The anti-ED1 antibody reacts against phagocytic macrophages, and we used the monoclonal antibody of ED1 (1:50, AbD Serotec, Oxford, UK) in this procedure. The frozen optic nerve sections were fixed and then blocked with 5% bovine serum albumin for 15 min. The primary antibody was applied and incubated overnight at 4 °C. The secondary antibody conjugated with fluorescein isothiocyanate (FITC, 1:100, Jackson ImmunoResearch Laboratories, West Grove, PA, USA) was incubated at room temperature for 1 hour. Counterstaining was performed using 4’,6-diamidino-2-phenylindole (DAPI, 1:1000, Sigma, St. Louis, MO, USA). For comparisons, the ED1-positive cells were counted in six HPFs (400× magnification) at the optic nerve lesion site.

### 4.8. Western Blotting and SOD Activity Assay

The retina samples were collected on day seven post-rAION induction in each group. The retina protein extracts were separated using a 4–12% NuPAGE Bis-Tris gel (Invitrogen, Carlsbad, CA, USA). The separated proteins were then transferred onto polyvinylidene difluoride membranes and blocked with 5% of nonfat milk in Tris buffer saline/Tween-20 solution containing 20 mM Tris-HCl (pH 7.5), 0.5 M NaCl and 0.5% Tween-20. The membranes were then incubated with p62, p-mTOR, p-AKT, GAPDH, pS6K (cell signaling technology, Inc, MA, US), Nrf2, IL-1β, TNFα, and SGK1 immunoglobulin (Abcam, Cambridge, MA, USA). The blots were then developed using Enhanced Chemiluminescent Substrate (Perkin-Elmer Life Science, Boston, MA, USA), and the relative intensities of the bands were measured using ImageJ software. The superoxide dismutase activity was measured according to the manufacturer’s protocol (Abcam, Cambridge, MA, USA). In brief, an enzyme mix was prepared followed by a standard curve preparation. Samples were loaded in 96 well plates in optimal dilutions and in triplicates. Plates were then incubated at 37 °C for 20 min. Optic density (OD) was measured at 450 nm and relative concentration was calculated in ug/ml. SOD activity was normalized to the sham group and the arbitrary units were plotted on a graph.

### 4.9. Statistical Analysis and Image Editing

All statistical analyses were performed using a GraphPad Prism software package. Mann–Whitney U test was used to compare the FVEP profiles and RGC’s densities between sham and rAION groups to validate the rAION induction and for all the experiments, Kruskal–Wallis test was used for comparisons among multiple groups, followed by Dunn’s multiple comparisons. *p* values less than 0.05 were considered statistically significant. All data are represented as mean ± standard deviation. The graphical abstract was created using an online platform biorender. Adobe Photoshop and Microsoft PowerPoint were used to generate the final images.

## Figures and Tables

**Figure 1 marinedrugs-18-00085-f001:**
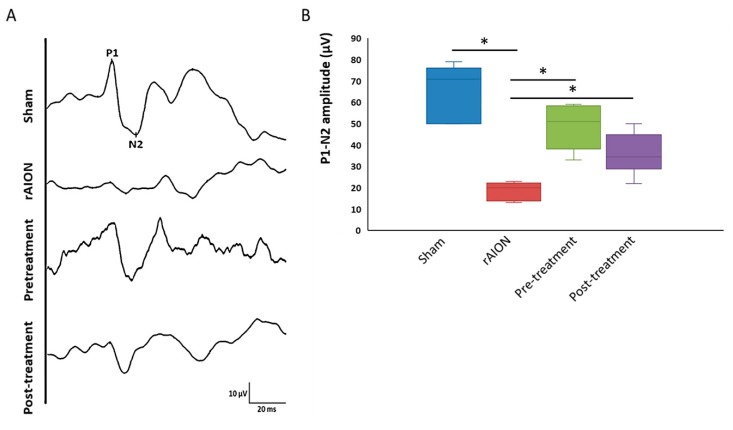
Astaxanthin preserves the visual function in rat model of anterior ischemic optic neuropathy (rAION). (**A**) representative flash visual-evoked potentials (FVEP) profiles of all groups. (**B**) The quantitative evaluation of visual function by measuring the P1-N2 amplitude. Both pre-and post-treatment showed a significant increase in P1N2 amplitude. *, *p* < 0.05, post hoc Dunn’s multiple comparisons test.

**Figure 2 marinedrugs-18-00085-f002:**
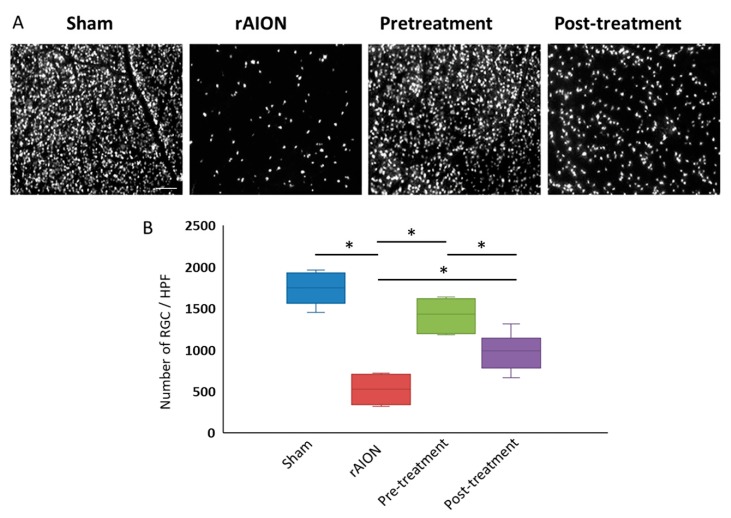
Astaxanthin treatment increases retinal ganglion cell (RGC) survival. (**A**) Representative image of fluoro-gold labeled RGCs in all groups. (**B**) Quantification of RGC density in the central retina, both pre- and post-treatment, had higher RGC density compared to rAION group. *, *p* < 0.05, post hoc Dunn’s multiple comparisons test; scale bar, 200 µm.

**Figure 3 marinedrugs-18-00085-f003:**
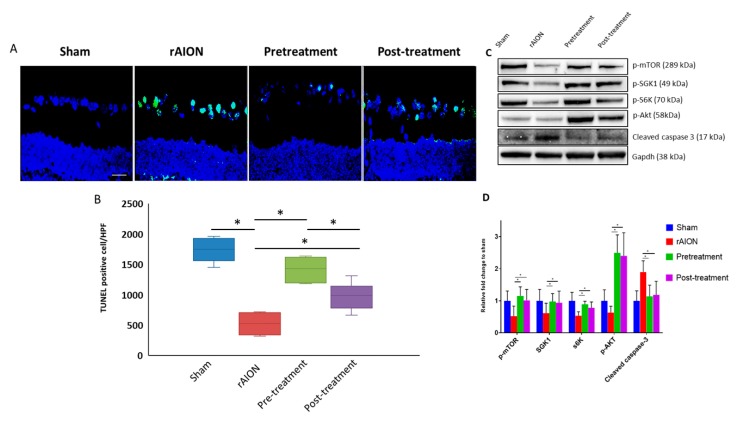
Astaxanthin treatment reduces apoptosis and provides an anti-apoptotic effect. (**A**) Representative images of terminal deoxynucleotidyl transferase dUTP nick end labeling (TUNEL) assay for all groups. (**B**) Quantification of TUNEL positive RGCs per high-powered field (cells/HPF) (green signal overlapping with 4′,6-diamidino-2-phenylindole (DAPI)). Pre- and post-treatment has fewer TUNEL positive RGCs compared to the rAION group. (**C**) Representative images of the immunoblot for all groups. (**D**) Quantification of immunoblots, both pre- and post-treatment, maintained levels of p-mTOR, p-S6K, and SGK1. Further, Akt activation, along with reduced levels of cleaved caspase-3 suggests the anti-apoptotic role of astaxanthin. *, *p* < 0.05, post hoc Dunn’s multiple comparisons test; scale bar, 100 µm.

**Figure 4 marinedrugs-18-00085-f004:**
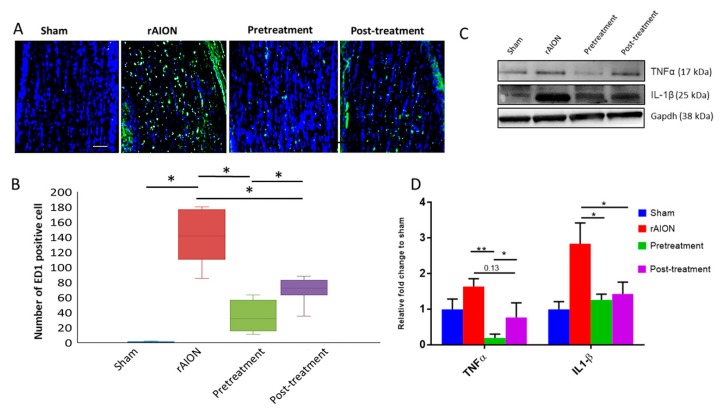
Astaxanthin exerts anti-inflammatory actions in the retina and halts macrophage infiltration in the optic nerve. (**A**) Representative images of ED1 stained (green) optic nerve longitudinal sections in different groups. (**B**) Quantification of ED1 positive cells. Both pre-and post-treatment reduced ED1 positive cells in the optic nerve compared to rAION. (**C**) Representative image of immune blots for retinal lysate in different groups. (**D**) Levels of TNFα and Il1β in the retina. The pre-treatment group has a better anti-inflammatory effect than post-treatment. *, *p* < 0.05; **, *p* < 0.01, post hoc Dunn’s multiple comparisons test; scale bar, 50 µm.

**Figure 5 marinedrugs-18-00085-f005:**
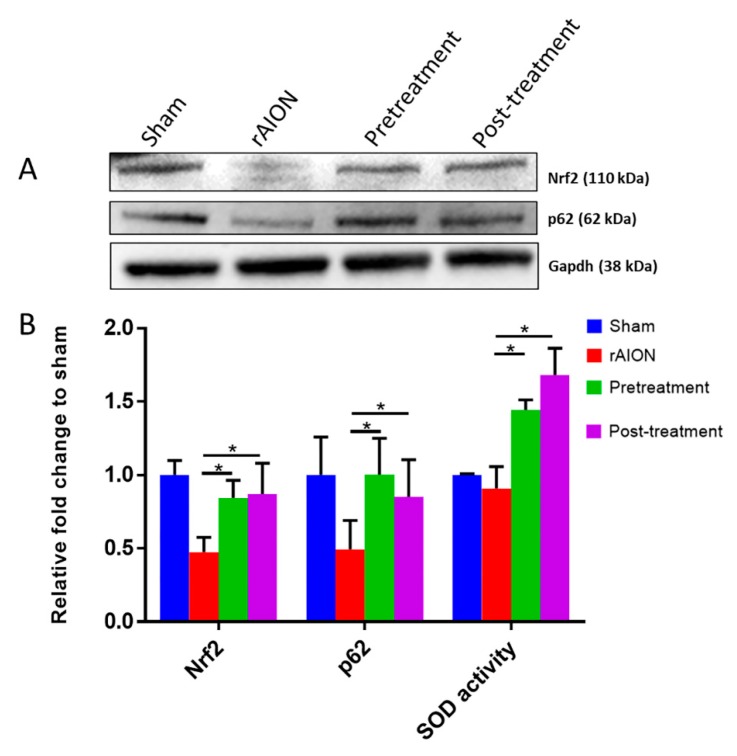
Anti-oxidative properties of astaxanthin. (**A**) Representative immunoblots of Nrf2 and p62. (**B**) Quantification of Nrf2, p62, and SOD activity in the retina. Both pre-and post-treatment showed anti-oxidative potential in the retina after ischemic stress. *, *p* < 0.05, post hoc Dunn’s multiple comparisons test.

**Figure 6 marinedrugs-18-00085-f006:**
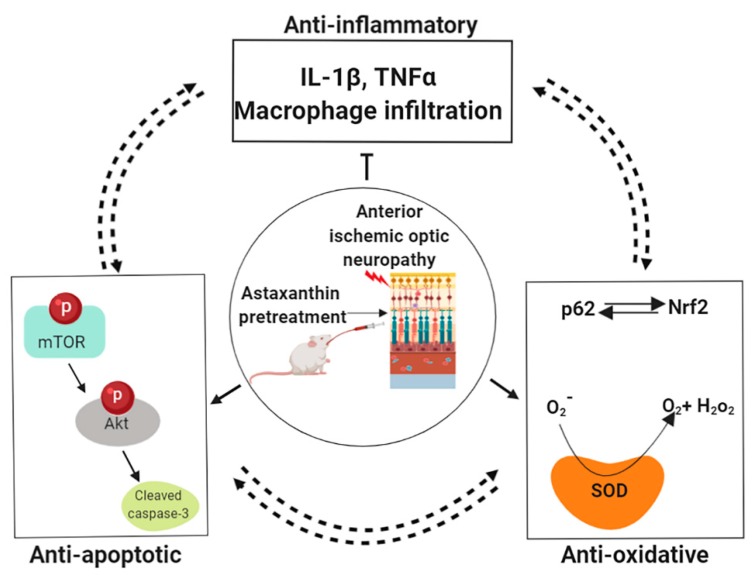
Summary of the neuroprotective effects of astaxanthin in rAION model. Astaxanthin pre-treatment provides a better anti-inflammatory, anti-oxidative, and anti-apoptotic compared to post-treatment.

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
