# Peer review of "Haematococcus pluvialis-Derived Astaxanthin Is a Potential Neuroprotective Agent against Optic Nerve Ischemia"

_marinedrugs, 2020, doi:10.3390/md18020085_

Round 1

Reviewer 1 Report

Comments for authors:

This study is established by clear data and authors demonstrated that pre-and post-treatment of astaxanthin can preserve visual function with reduce RGC apoptosis after ischemic insults involving antioxidative effect of higher expression of p-Akt, p-mTOR, Nrf2 and SOD. In addition, authors described astaxanthin of pretreatment provides a better anti-inflammatory, anti-oxidative, and anti-apoptotic compared to post-treatment (Figure 6). However, any results are not significant difference between the pre and post- treatment. Authors should reconsider and/or discuss about this thing.

Author Response

Point 1

This study is established by clear data and authors demonstrated that pre-and post-treatment of astaxanthin can preserve visual function with reduce RGC apoptosis after ischemic insults involving antioxidative effect of higher expression of p-Akt, p-mTOR, Nrf2 and SOD. In addition, authors described astaxanthin of pretreatment provides a better anti-inflammatory, anti-oxidative, and anti-apoptotic compared to post-treatment (Figure 6). However, any results are not significant difference between the pre and post- treatment. Authors should reconsider and/or discuss about this thing.

Response 1

Thank you for the reviewer's suggestion. We already performed the statistical analysis between Pre- and Post-treatment. We found significant changes (p<0.05) in

RGC density(figure2) Line 72-74

TUNEL assay (figure3) Line 86-87

ED1+ cells (figure 4) line 98-99.

However, there was no significant difference between groups in the VEP analysis (figure 1), (line 73-74) and we have included this information in our revised manuscript

Reviewer 2 Report

The present study from Wei-Ning Lin et al. is of potential interest, written in a quite appropriate way and adequately conducted. The manuscript reports that treatment of adult male Wistar rats with astaxanthin (100mg/kg/day) for 7 consecutive days either before or after inducing oxidative stress in the retina by photodynamic therapy increases the expression of p-Akt, p-mTOR, Nrf2 and superoxide dismutase activity, and reduce cleaved caspase-3 expression, optic nerve macrophage infiltration  and RGC apoptosis.

All conclusions are justified and supported by the results. The data and analyses are presented appropriately and the techniques are appropriate and well conducted. However, Authors oscillate among the examination of the retinas and optic nerve and do not seem to find a stong hypothesis to explain the action fo astaxanthin they find equaly effective in both retinas and optic nerve. Flash Visual-Evoked Potentials and the labelling of the RGCs through the superior colliculi are correct, but more attention to the consequentiality of the findings should be paid in the results.

Undoubtedly, there is increasing interest in the use of astaxanthin as an antioxidant able to prevent oxidative stress in turn involved in the pathogenesis and progression of age-related neurodegenerative diseases. Astaxanthin was shown to improve mitochondrial integrity. Mitochondrial damage and oxidative stress are major players in all of the retinal diseases.

Reactive Oxygen Species (ROS) are generated from either mitochondrial or nonmitochondrial sources, including NADPH, xanthine oxidase and others, but Authors appear to ignore that a major ROS source is the extra-mitochondrial oxidative phosphorylation that occurs in the rod Outer Segments and in myelin (doi: 10.1021/acs.jproteome.7b00741. https://doi.org/10.1007/s12035-015-9216-0).The ectopic respiratory chain would be a primary unshielded source of Reactive Oxygen Species. An extra-mitochondrial oxidative phosphorylation was reported also in many cellular membranes (see doi: 10.1016/j.bbabio.2008.08.003.; Shimizu N, et al. Am J Physiol Heart Circ Physiol. 2007 Sep;293(3):H1646-53), and also in platelets (doi: 10.1111/boc.201700025.) and extracellular vesicles (DOI: 10.1080/14789450.2018.1528149.

Recent evidence shows that the primary retinal damage from oxidative stress production occurs to rod outer segments and not only in the inner segment containing the mitochondria, but also in the Outer limb (see the data from Funk, Roelke et al.). Consistently, Authors themselves found that “oral supplementation of astaxanthin significantly reduces retinal damage and alleviates the decrease in electroretinogram a- and b- wave amplitudes following retinal ischemia and reperfusion.[15]” This also points to a similar role played by astaxanthin in both retina and optic nerve which are quite different as for structure and function.

The work could provide an advance towards the current knowledge, if it discussed the interesting data in light of the new findings of a source of oxidative stress inside the photoreceptor outer segments and optic nerves, besides mitochondria.

In general, the cited literature is in general not sufficiently up-to date, fundamental literature that would be pivotal to cite is missing: for example specifically, the paper appeared on this Journal Mar. Drugs 201816(8), 247; https://doi.org/10.3390/md16080247 and https://doi.org/10.1155/2019/3849692  should be cited.  English language can be improved, here are some examples:

Line 19 “very less” should be: much less

Line 20: “optic nerve  ischemia” should sound: ischemic optic nerve  

Line 2..”prevent oxidative stress-induced  retinal ganglion cell (RGC) death by anti-oxidative and anti-apoptotic responses” is clumsy.

Line 2..”:  The adult “ should simply be: Adult

Line 58 ..anti-cancer effects in the literature should be: in Literature

Author Response

Point 1

All conclusions are justified and supported by the results. The data and analyses are presented appropriately and the techniques are appropriate and well conducted. However, Authors oscillate among the examination of the retinas and optic nerve and do not seem to find a strong hypothesis to explain the action of astaxanthin they find equally effective in both retinas and optic nerve.

Response 1

We want to comment that AION mainly affects optic nerve, which triggers inflammatory cascade in the optic nerve and later leading to retrograde axon degeneration leading to RGC death. Therefore, we need evidences both from optic nerve and retina to make our findings more convincing and potential. Hence, VEP, RGC counting and TUNEL were performed to check RGC survival and visual function respectively, and ED1 staining to verify the macrophage/microglial activation and inflammation in the optic nerve. We have emphasized the hypothesis in the end of discussion (line 198-200)

Point 2

Flash Visual-Evoked Potentials and the labelling of the RGCs through the superior colliculi are correct, but more attention to the consequentiality of the findings should be paid in the results.

Response 2

We have added a brief description of what to infer from VEP (line 68-69) and Flour-gold labeling of RGC’s (line 77-78).

Point 3

Undoubtedly, there is increasing interest in the use of astaxanthin as an antioxidant able to prevent oxidative stress in turn involved in the pathogenesis and progression of age-related neurodegenerative diseases. Astaxanthin was shown to improve mitochondrial integrity. Mitochondrial damage and oxidative stress are major players in all of the retinal diseases. Reactive Oxygen Species (ROS) are generated from either mitochondrial or nonmitochondrial sources, including NADPH, xanthine oxidase and others, but Authors appear to ignore that a major ROS source is the extra-mitochondrial oxidative phosphorylation that occurs in the rod Outer Segments and in myelin (doi: 10.1021/acs.jproteome.7b00741. https://doi.org/10.1007/s12035-015-9216-0).The ectopic respiratory chain would be a primary unshielded source of Reactive Oxygen Species. An extra-mitochondrial oxidative phosphorylation was reported also in many cellular membranes (see doi: 10.1016/j.bbabio.2008.08.003.; Shimizu N, et al. Am J Physiol Heart Circ Physiol. 2007 Sep;293(3):H1646-53), and also in platelets (doi: 10.1111/boc.201700025.) and extracellular vesicles (DOI: 10.1080/14789450.2018.1528149. Recent evidence shows that the primary retinal damage from oxidative stress production occurs to rod outer segments and not only in the inner segment containing the mitochondria, but also in the Outer limb (see the data from Funk, Roelke et al.). Consistently, Authors themselves found that “oral supplementation of astaxanthin significantly reduces retinal damage and alleviates the decrease in electroretinogram a- and b- wave amplitudes following retinal ischemia and reperfusion.[15]” This also points to a similar role played by astaxanthin in both retina and optic nerve which are quite different as for structure and function.The work could provide an advance towards the current knowledge, if it discussed the interesting data in light of the new findings of a source of oxidative stress inside the photoreceptor outer segments and optic nerves, besides mitochondria.

In general, the cited literature is in general not sufficiently up-to date, fundamental literature that would be pivotal to cite is missing: for example specifically, the paper appeared on this Journal Mar. Drugs 2018, 16(8), 247; https://doi.org/10.3390/md16080247 and https://doi.org/10.1155/2019/3849692  should be cited.

Response 3

We have referred the above mentioned articles that the reviewer has indicated and we agree with the reviewer’s opinion that ROS occurs non-mitochondrial sources like in rod outer segment and myelin. But the animal model used in our study does not affect the outer segment. We used a green laser to generate ROS only in the optic nerve head, primarily, this it affects the optic nerve head and then due to the inflammatory cascade, retrograde axon degeneration occurs leading to RGC death. Further, we found no apoptosis in outer segment checked TUNEL assay.

We have also updated our manuscript with citations mentioned above (line 57) (reference 15 and 16).

Point 4

English language can be improved, here are some examples:

Line 19 “very less” should be: much less

Line 20: “optic nerve  ischemia” should sound: ischemic optic nerve 

Line 2..”prevent oxidative stress-induced  retinal ganglion cell (RGC) death by anti-oxidative and anti-apoptotic responses” is clumsy.

Line 2..”:  The adult “ should simply be: Adult

Line 58 ..anti-cancer effects in the literature should be: in Literature

Response 4

We appreciate for reviewers detailed comments and proofreading. And we have corrected our manuscript accordingly.

Reviewer 3 Report

The effect of oral astaxanthin on the optic nerve damage due to putative AION was persuasively studied using a rat model of AION (rAION). In order to improve the manuscript, I would suggest some minor points.

Introduction and Discussion: In addition to the general predisposing factors including hypertension, diabetes mellitus, hyperlipidemia, nocturnal hypotension, smoking and sleep apnea, small optic disc is an important local anatomic risk factor  for AION (Hayreh SS, Prog Retin Eye Res, 2016). Particularly, if astaxanthin was effective, to some extent, to prevent  the optic nerve damage due to AION, it would be nice for people with small optic discs (disc at risk) to have astaxanthin for preventive purpose. The validity of rAION to simulate human AION needs to be discussed as the limitation of this study, as the anatomy of the optic disc ia somewhat different between rats and humans.

Author Response

Review 3

Point 1

Introduction and Discussion: In addition to the general predisposing factors including hypertension, diabetes mellitus, hyperlipidemia, nocturnal hypotension, smoking and sleep apnea, small optic disc is an important local anatomic risk factor for AION (Hayreh SS, Prog Retin Eye Res, 2016).

Response 1

Thanks for your suggestion, we have added this factor and reference in our revised manuscript. Line 41, reference 4

Point 2

Particularly, if astaxanthin was effective, to some extent, to prevent the optic nerve damage due to AION, it would be nice for people with small optic discs (disc at risk) to have astaxanthin for preventive purpose.

Response 2

We agree with the reviewer that astaxanthin supplementation can be beneficial for people with disc at risk. We have added a comment in our revised manuscript in discussion section. (Line 200)

Point 3

The validity of rAION to simulate human AION needs to be discussed as the limitation of this study, as the anatomy of the optic disc is somewhat different between rats and humans.

Response 3

The major and critical similarities in rodent AION and Clinical NAION are

Optic nerve head edema Loss of Retinal ganglion cells No latency in VEP responses and only amplitude changes in affected eyes Inflammatory immune cell infiltration, microglia activation was also found in clinical NAION which are similar to rodent model and primate model of NAION. https://jamanetwork.com/journals/jamaophthalmology/fullarticle/1106510 http://iovs.arvojournals.org/article.aspx?articleid=2127587

However, the limitation of this study is that the animals used were young and without any predisposing factors like diabetes hypertension sleep apnea that we see as risk factors of NAION in humans.

We have cited the above mentioned points and articles in our manuscript as well. Line (192-196). Reference (37-39)

Reviewer 4 Report

This manuscript is potentially interesting, but I found may serious problems:

The number of animals is low and, in some cases, not acceptable for a reliable statistical analysis, as in the case of experiments based on n=4. The statistical analysis is badly reported in results, as authors never mentioned the statistical test used for comparisons. It is also unclear how data were reported. As the authors used non-parametric test, they should illustrate findings by using boxplots. In methods, authors misinterpreted the statistical analysis, as they stated to have used the Kruskal-Wallis test for group comparisons. This is certainly not true since the Kruskal-Wallis test is the counterpart of analysis of variance used in the case of non-normal distribution. Thus, authors must re-analyze their findings making comparisons by the Dunn’s test. Description for superoxide dismutase activity analysis is missing. The authors may refer to papers describing the standard test, such as Biagini et al. 1995 (doi: 10.1016/0304-3940(95)11529-6) or Marzatico et al. 1998 (doi:10.1023/A:1022969828885). Note that immersion in 30% sucrose is used for cryoprotection, not for dehydration. There is no mention on how rats were killed and of the anesthesia. The reduction of TNF alpha levels by pretreatment is very surprising and should be explained. In the abstract, correct photodynamic therapy (treatment).

Author Response

Point 1

The number of animals is low and, in some cases, not acceptable for a reliable statistical analysis, as in the case of experiments based on n=4. The statistical analysis is badly reported in results, as authors never mentioned the statistical test used for comparisons

Response 1

In our current manuscript sample size of n=4 is only for western blotting analysis from different biological replicate. The experiment was performed three times using each biological replicate.

Other tests like VEP, Flouro-gold, TUNEL, and ED1 we used n=6. Based on our preliminary data and our previously published articles, we performed sample size estimation using G*Power. Our statistical analysis suggests that the minimum sample size in each group among four groups (Sham, rAION, Pretreatment and post-treatment) in VEP, Flouro-gold, TUNEL, ED1 and Western blotting analysis with 95% power is 4, 4, 3, 3 and 3 respectively. Further, we used adult male Wistar rats- the size of their eyes is big enough that AION induction can be accurately performed by experienced operators when compared to mice, where the sample size should be more abundant to reduce the standard deviation. We have also added a short description in methods section titled “dosage information and sample size estimation for treatment group “section. Line (245-250), reference 45

Figure1- Screen shot of Power analysis for sample size estimatioons for Flouro-gold labelling of RGC’s. Note the red boxes indication power of the test, minimum total sample size and the Standatd deviation within groups. Similarly we have performed this test to calculate the sample size for different experiemtns mentioned above in Response 1

Figure 2- X-Y plot of sample size vs power of the test. Note that after Total sample size of 20 the experiment attains maximum power. We used total sample size of 24.

Even though we expect an effect at 16, we used total sample size of 24.  This will help ensure that we have enough power in case some of the pre and post-treatment assumptions mentioned above are not met. The authors hope that this carefully answers reviewer’s concerns.

Point 2

It is also unclear how data were reported. As the authors used non-parametric test, they should illustrate findings by using boxplots.

Response 2

We have changed the graph style from bar to boxplots in X-Y charts. However, in western blotting grouped charts it will be easier for reader eyes to infer from the bar graphs due to its smaller size. Therefore, all the authors request to keep the western blotting charts without any changes.

Point 3

In methods, authors misinterpreted the statistical analysis, as they stated to have used the Kruskal-Wallis test for group comparisons. This is certainly not true since the Kruskal-Wallis test is the counterpart of analysis of variance used in the case of non-normal distribution. Thus, authors must re-analyze their findings making comparisons by the Dunn’s test.

We used GraphPad prism to perform the statistical analysis. And in this version of graph pad, Dunn’s test is automatically performed under the Kruskal Wallis test. All the p-values for multiple Comparisons are from Dunn's test. We have included a description in our revised manuscript’s methods section of how the tests were performed (line 295). We have provided the screenshot of one of the tests for better understanding.

Figure 3 A sample display of the graphpad Prism software. Note that Kruskal Wallis and Dunn’s test is performed at the same time.

Point 4

Description for superoxide dismutase activity analysis is missing.

Response 4

The SOD activity was performed according to the manufacturer’s protocol and It is included in Western blotting and SOD activity assay section.

Point 5

Note that immersion in 30% sucrose is used for cryoprotection, not for dehydration. Line 65

Response 5

We have made the necessary correction

Point 6

There is no mention on how rats were killed and of the anesthesia.

Response 6

We have added a small description about animal anesthesia and euthanasia in our revised manuscript Line (214 and 221)

Point 7

The reduction of TNF alpha levels by pretreatment is very surprising and should be explained.

Response 7

We have discussed this in our revised manuscript. Line (186-188)

Our hypothesis is that in pretreatment group the residual astaxanthin in body may have depleted the ROS already after rAION induction. Resulting in very strong anti-inflammatory effect compared to post treatment.

Point 8

In the abstract, correct photodynamic therapy (treatment). 

Response 8

We have corrected the term in our revised manuscript. Line (23)

Round 2

Reviewer 4 Report

Authors did not adequately modified the manuscript. Statistical results are still incompletely described, since the p value is not accompanied by detailing the statistical procedure followed to obtain it. Whart does it mean the sentence "The number of TUNEL-positive cells in the RGC layer of the sham, rAION, pretreatment, and post-treatment group retinas were 1.4±0.5, 12.4±2.2, 3.6±1.9, and 6.2±3.3, respectively"? I see much more cells in the mentioned figure. Additionally, authors did not add the required references to techniques' description.

Author Response

Point 1

Authors did not adequately modified the manuscript. Statistical results are still incompletely described, since the p value is not accompanied by detailing the statistical procedure followed to obtain it.

Response 1

We have now clearly described the statistical analysis used along with the procedure used to obtain the p-value. Line 300-303

And the test used is also mentioned in each figure legend

Figure1 Line 115-116

Figure2 Line 122

Figure3 Line 131-132

Figure4 Line 141

Figure5 Line 145-146

Point 2

Whart does it mean the sentence "The number of TUNEL-positive cells in the RGC layer of the sham, rAION, pretreatment, and post-treatment group retinas were 1.4±0.5, 12.4±2.2, 3.6±1.9, and 6.2±3.3, respectively. I see much more cells in the mentioned figure.?

Response 2

TUNEL positive cells in this context means TUNEL positive RGCs, we have replaced the words from ‘TUNEL positive cells’ to ‘TUNEL positive RGCs’ in our revised manuscript for better understanding. And, we have provided a brief description in methods section about how the analysis was carried out (Line 276-278). We concluded TUNEL positive RGCs as apoptotic RGCs only if the TUNEL (green color) overlapped with DAPI (Blue). Only Green signal with no DAPI was considered as false positive and were not counted as TUNEL positive RGCs. And the values mentioned above are the average number of TUNEL positive RGCs.

Point 3

Additionally, authors did not add the required references to techniques' description.

Response 3

For further clarification, we have added the reference from which the TUNEL assay kit was developed (Line 271) Reference 52

Round 3

Reviewer 4 Report

Authors partially accomplished the required corrections. Description of statistical results is still missing in the main text illustrating results. For instance, in line 52 the p value is not followed by description of the statistical test used to get it. Similarly, in other paragraphs of results. In methods, the required references for SOD activity analysis are still missing. Also indicate units for measurements. For instance, in line 96 (and others) this information is missing.

Author Response

Point 1

Authors partially accomplished the required corrections. Description of statistical results is still missing in the main text illustrating results. For instance, in line 52 the p value is not followed by description of the statistical test used to get it. Similarly, in other paragraphs of results.

Response 1

We have added the description of statistical test used in the text illustrating results in all the results section

Line 72, 83, 87, 94, 106, 108 and 115

Point 2

In methods, the required references for SOD activity analysis are still missing. Also indicate units for measurements. For instance, in line 96 (and others) this information is missing.

Response 2

The SOD activity was measured using a commercially available kit. We did not refer to any of the previously published articles to perform SOD assay. Therefore we have now described, in brief, how the SOD activity was carried out in our methods section Line 330-334. We have also added the units wherever necessary in out revised manuscript Line 333.

*The Line number reference in the response letter is according to the manuscript in PDF format.